# Risk Factors for Mortality in Emergently Admitted Patients with Acute Gastric Ulcer: An Analysis of 15,538 Patients in National Inpatient Sample, 2005–2014

**DOI:** 10.3390/ijerph192316263

**Published:** 2022-12-05

**Authors:** Maksat Idris, Abbas Smiley, Saral Patel, Rifat Latifi

**Affiliations:** 1New York Medical College, School of Medicine and Westchester Medical Center, Valhalla, NY 10595, USA; 2Department of Surgery, University of Arizona, Tucson, AZ 85721, USA

**Keywords:** acute gastric ulcer, emergency admission, surgery, hospital length of stay, frailty

## Abstract

Background: Patients admitted emergently with a primary diagnosis of acute gastric ulcer have significant complications including morbidity and mortality. The objective of this study was to assess the risk factors of mortality including the role of surgery in gastric ulcers. Methods: Adult (18–64-year-old) and elderly (≥65-year-old) patients admitted emergently with hemorrhagic and/or perforated gastric ulcers, were analyzed using the National Inpatient Sample database, 2005–2014. Demographics, various clinical data, and associated comorbidities were collected. A stratified analysis was combined with a multivariable logistic regression model to assess predictors of mortality. Results: Our study analyzed a total of 15,538 patients, split independently into two age groups: 6338 adult patients and 9200 elderly patients. The mean age (SD) was 50.42 (10.65) in adult males vs. 51.10 (10.35) in adult females (*p* < 0.05). The mean age (SD) was 76.72 (7.50) in elderly males vs. 79.03 (7.80) in elderly females (*p* < 0.001). The percentage of total deceased adults was 1.9% and the percentage of total deceased elderly was 3.7%, a difference by a factor of 1.94. Out of 3283 adult patients who underwent surgery, 32.1% had perforated non-hemorrhagic ulcers vs. 1.8% in the non-surgical counterparts (*p* < 0.001). In the 4181 elderly surgical patients, 18.1% had perforated non-hemorrhagic ulcers vs. 1.2% in the non-surgical counterparts (*p* < 0.001). In adult patients managed surgically, 2.6% were deceased, while in elderly patients managed surgically, 5.5% were deceased. The mortality of non-surgical counterparts in both age groups were lower (*p* < 0.001). The multivariable logistic regression model for adult patients electing surgery found delayed surgery, frailty, and the presence of perforations to be the main risk factors for mortality. In the regression model for elderly surgical patients, delayed surgery, frailty, presence of perforations, the male sex, and age were the main risk factors for mortality. In contrast, the regression model for adult patients with no surgery found hospital length of stay to be the main risk factor for mortality, whereas invasive diagnostic procedures were protective. In elderly non-surgical patients, hospital length of stay, presence of perforations, age, and frailty were the main risk factors for mortality, while invasive diagnostic procedures were protective. The following comorbidities were associated with gastric ulcers: alcohol abuse, deficiency anemias, chronic blood loss, chronic heart failure, chronic pulmonary disease, hypertension, fluid/electrolyte disorders, uncomplicated diabetes, and renal failure. Conclusions: The odds of mortality in emergently admitted geriatric patients with acute gastric ulcer was two times that in adult patients. Surgery was a protective factor for patients admitted emergently with gastric perforated non-hemorrhagic ulcers.

## 1. Introduction

A gastric ulcer is a very serious condition that has many different presentations. Most of the literature refers to gastric ulcers under the umbrella of peptic ulcers. In general, peptic ulcers include both gastric and duodenal ulcers, although the pathophysiology of the two have differences [1]. These differences are usually typified by the site and route of infection—whether it is a *H. pylori* infection or an overuse of NSAIDS—and the differing chemical environments within the two lumens [1]. Despite these differences, data from studies looking at peptic ulcers can be used to make inferences about gastric ulcers, as much of the literature has been doing so.

Perforated gastric ulcers are of particular concern in patients with ulcers. Perforated gastric ulcers come with a two- to three-times greater risk of death overall and four-times greater risk postoperatively than duodenal ulcers [1,2] Historically, perforations in the upper gastrointestinal tract have always been lethal, with mortality rates ranging from 40 to 60% [3]. However, with the introduction of antibiotics and general improvements in surgical techniques over the years, these numbers have decreased, and patients have had better outcomes [4]. Despite these changes in treatment approaches, perforations still carry a great risk for mortality, and patients need an individualized approach to treatment, with an early diagnosis and early treatment being important factors in the discussion [5,6]. Although surgery is not necessary in some situations [7,8], multiple authors have agreed that surgical treatment is necessary in situations such as ulcers increasing in size, failing to heal completely, perforations, and especially malignancies [9,10,11]. Interestingly, unlike duodenal ulcers, which have relatively insignificant rates of cancer, gastric ulcers are usually correlated with malignancies, which contribute to its increased morbidity and mortality [11,12]. Thankfully, due to new diagnostic procedures, such as endoscopic examination and a biopsy follow up, clinicians have had success at detecting gastric ulcers, as well as ruling out any malignancies [2,9].

Many risk factors contribute to the mortality of patients with gastric ulcers, with perforations being especially dangerous. The aim of this study was to find and analyze these risk factors and determine if there was any age-related difference in the mortality rate of emergently admitted patients with acute gastric ulcer using the National Inpatient Sample database from 2005 to 2014.

## 2. Materials and Methods

We used the National Inpatient Sample database from 2005 to 2014, which is a large and publicly available inpatient care database across the United States, with data on more than 70 million hospital stays. Our study looked at 15,538 patients, between 2005 and 2014, with emergently admitted acute gastric ulcers. Using this database, we performed a retrospective study on a cohort of patients with acute gastric ulcers, separated into two age groups: adult (ages 18–65 years) and elderly (ages 65+ years). ICD-9 codes used to identify the patients used for this study were 531.0, 531.1, 531.2. The following variables were collected for analysis: age, gender, gastric ulcer type, invasive diagnostic procedures, surgical procedure status, survival, modified frailty index, time to surgical procedure (time to surgery), hospital length of stay (HLOS), and associated comorbidities. The modified frailty index was calculated using: (1) the presence of at least 1 of the following: solid tumor, renal failure, metastatic cancer, paralysis, lymphoma, coagulopathy, and weight loss; (2) diabetes, complicated or uncomplicated; (3) chronic pulmonary disease; (4) congestive heart failure; (5) hypertension (un/complicated). Functional health status was not available in the dataset and was estimated using available variables. If the patient was comorbid for: tumor, renal failure, metastatic cancer, paralysis, lymphoma, coagulopathy, or weight loss, then it could be assumed that the patient was functionally dependent (partial or total dependence).

For the statistical analysis, descriptive and analytical statistics were used. Mean, standard deviation (SD), and confidence interval (CI) at 95% were calculated when dealing with numerical values. Categorical values were compared using chi-square analysis, and continuous variables were compared using a two-sample standard t-test, which used continuous variables (in our case) to test whether the mean values of two populations/variables were statistically different from one another. A multivariable logistic regression model with backward elimination was used to evaluate the most significant factors related to mortality, eliminating the least significant factors in a stepwise fashion. After elimination, the following variables remained: age, sex, time to surgery, modified frailty index, ulcer type, and HLOS and invasive diagnostic procedures. *p* values less than 0.05 were considered statistically significant, and analyses were performed using SPSS software version 24 (SPSS Inc., Chicago, IL, USA) and R statistical software (Foundation for Statistical Computing, Vienna, Austria).

## 3. Results

There was a total of 15,538 patients included in this study, which consisted of 6338 adult patients and 9200 elderly patients. The mean age (SD) for adult males was 50.42 (10.65) years compared to 51.10 (10.35) years in adult females. The mean age (SD) for elderly males was 76.72 (7.50) years compared to 79.03 (7.80) years in elderly females (Table 1). In both age groups, males were more likely to undergo surgical procedures. For both age groups, females experienced a higher percentage of perforation, while males experienced a higher percentage of hemorrhage. Generally, mortality was similar between both sex categories in both age groups as shown through an insignificant *p*-value (Table 1).

The following comorbidities were associated with at least 15% of cases in one or both age groups: alcohol abuse, deficiency anemias, chronic blood loss, congestive heart failure, chronic pulmonary disease, uncomplicated diabetes, hypertension, fluid electrolyte disorders, and renal failure (Table 1). Of these comorbidities, the ones with a female bias included: chronic pulmonary disease (adult), hypertension (elderly), and fluid/electrolyte disorders (both). Comorbidities with a male bias included: alcohol abuse (both), chronic pulmonary disease (elderly), uncomplicated diabetes (both), and renal failure (both).

Of these comorbidities, only two were associated with about 30% of cases: hypertension (42.3% for adults and 68.3% for elderly) and fluid/electrolyte disorders (29% for adults and 33% for elderly). Although associations were less than 15% of cases, lymphomas, metastatic cancer, and solid tumors were included due to findings in the literature that point to associations with cancer, which is further explored in the discussion. Of these, only solid tumors had a male bias in the elderly population.

### 3.1. Age and Ulcer Type and Their Relationship with Mortality

Overall, age contributed to mortality, demonstrated by increased mortality in the elderly within each ulcer group as evident in Figure 1. Table 2 values indicate that mortality in all elderly patients was 2× greater than in all adult patients. Additionally, the average age was higher in the deceased group compared to the survived group for both age populations (Table 2). Hemorrhagic ulcer patients experienced the lowest mortality when compared to other ulcer types (perforated ulcer or hemorrhage perforated ulcer) within the same age group. In both age groups, hemorrhagic ulcer patients had the lowest mortality, while hemorrhagic perforated ulcer patients had the highest mortality in adults, and perforated ulcer patients had the highest in the elderly (Figure 1).

### 3.2. The Role of Surgery

When surgical (w/Surgery) and nonsurgical (w/o Surgery) groups were compared, a significantly lower mortality percentage was found in nonsurgical groups, which aligns with the accepted theory of increased morbidity/mortality secondary to surgery (Figure 2).

Additionally, most perforated ulcer and hemorrhagic perforated ulcer patients elected to have surgery, while a larger percentage of hemorrhagic ulcer patients did not elect to have surgery. Furthermore, when assessing percentages within Table 3, the proportion of perforated ulcer adults with operations was about 18× greater than perforated ulcer adults without operation, which is an index of severity. In direct comparison, the mortality rate (%) increased by a factor of 2.4 when comparing adults with surgery to those without. The lack of an 18-fold increase in mortality rate (%) between the two groups indicates surgery as a protective factor for this cohort of perforated ulcer patients. Similarly, the percentage of elderly patients with perforated ulcer was 15× higher in those who had surgery compared to those who did not; mortality is increased by a factor of 2.5 when comparing the percent deceased in the elderly operative group to the non-operative group.

### 3.3. Time to Surgery

Based on Figure 3B, Figure 4B and Table 4, the time to surgery for perforated ulcer patients was shortest in both age groups (*p* < 0.05). Table 4 compares risk factors amongst the different ulcer groups. For both age groups, patients who had a perforation (perforated ulcer and hemorrhagic perforated ulcer) experienced longer HLOS, longer time to invasive diagnostic procedure, and higher rate of surgical procedures. Patients with hemorrhage (hemorrhagic ulcer) experienced more invasive diagnostic procedures for both age groups.

Within the adult cohort, the time (days) it took to get to surgery (time to surgery) was significantly lower in the group of adults that survived when compared with the group of adults that were deceased (Figure 3A). The elderly group, on the other hand, had a non-significant difference in time to surgery between survived and deceased patients (Figure 4A).

### 3.4. Mortality and Hospital Length of Stay (HLOS)

The number of days spent in the hospital for patients in this study was significantly lower for those who survived than those who did not (Figure 3A, Figure 4A and Table 2). Furthermore, when comparing HLOS from Figure 3B and Figure 4B and overall mortality in Figure 1, a similar pattern emerged: there was an overall correlation between mortality and HLOS, such that both variables shared similar patterns of stratification based on ulcer type. In other words, larger HLOS values (days) were correlated with increased risks for mortality, and smaller HLOS values (days) were correlated with lower risks for mortality. For example, elderly perforated ulcer patients had the largest HLOS value as compared to hemorrhagic ulcer and hemorrhagic perforated ulcer patients (Figure 3B), while elderly perforated ulcer patients also experienced the greatest mortality percentage when compared to mortality in the hemorrhagic ulcer and hemorrhagic perforated ulcer patients (Figure 1).

### 3.5. Risk Factors for Mortality

Of those in the adult group who underwent surgery, the following factors were considered in the multivariable logistic regression models (prior to backward elimination): time to surgery, modified frailty index, ulcer type, age, sex, invasive diagnostic procedures, race, income, insurance, and hospital location. Table 5 and Table 6 indicated age as a significant contributing factor to mortality within elderly surgical and nonsurgical groups. Age was used as a continuous variable in this study, which allowed for a deeper understanding between the relationship of age and mortality. This showed that for the elderly population, there is about a 5% and 4% increase in mortality for operative and non-operative patients, respectively, with each additional year of age. Time to surgery and modified frailty index had a positive relationship with mortality such that increases in either time to surgery or modified frailty index led to increased risk for mortality in those surgical patients (Table 5). Regarding ulcer type, both perforated ulcer and hemorrhagic perforated ulcer adults had strong positive relationships with mortality (Table 5). Table 5 found significant positive odds ratios between mortality and perforated ulcer, and mortality and hemorrhagic perforated ulcer in both adult and elderly surgical groups, further confirming the effect of perforations on mortality.

Within the elderly surgical group, the following factors were considered in the multivariable logistic regression models (prior to backward elimination): time to surgery, modified frailty index, ulcer type, age, sex, invasive diagnostic procedures, race, income, insurance, and hospital location. (Table 5). Time to surgery, modified frailty index, ulcer type, *and* age shared positive relationships with mortality. In contrast, there was a negative relationship between female sex and mortality, indicating that elderly males with gastric ulcers were at a higher risk for mortality than elderly females with the same condition.

When considering adults and elderly that did *not* undergo surgery (Table 6), invasive diagnostic procedures reduced the odds of mortality, whereas longer stay in hospital increased the odds of mortality. When considering only elderly nonsurgical groups, additional positive odds ratios were found between odds of mortality and ulcer type, age, and modified frailty index.

### 3.6. Comorbidities and Secondary Diagnoses

Table 7 lists secondary diagnoses of the patients in this study, stratified according to survival status. Secondary diagnoses that make up at least 50% of patients in the survived and deceased categories (either age group) are as follows: hypertension, anemia and/or hemorrhage, cardiac diseases, diseases of the digestive system other than the liver, and genitourinary system diseases. The figures of 15% and 50% were arbitrary numbers picked relative to other comorbidities and secondary diagnoses, but it aids in narrowing down those diseases from the total list of all associated comorbidities and secondary diagnoses, as some were as low as 0.2% and 0%, respectively.

## 4. Discussion

### 4.1. Age and Mortality

A gastric ulcer can be a very serious condition, and a variety of factors influence mortality in these patients. It is commonly accepted, and reinforced in the literature, that the elderly population is at a significant risk for worse outcomes and mortality when faced with dire medical situations [13,14,15]. These age-related trends stand in the case of gastric ulcers, particularly those that are perforated [6,16]. The current study corroborates these findings with values that indicate elderly patients are at a 2-fold greater risk for mortality than adult patients. Additionally, a backward linear regression model found similar increases in mortality within elderly patients when considering age as a factor. When looking at overall trends in perforated peptic ulcers between 1935 and 1990, the lowest mortality was between 1950 and 1980, but it steadily increased as the years approached 1990. This tail end increase in death rate was due to the increase in the percentage of elderly patients with perforated gastric ulcers [4]. Although the adult group in the current study had more patients with perforated gastric ulcers than the elderly group, the elderly group with perforated gastric ulcers had significantly higher death rates. This exemplifies the contribution to mortality in recent years from the elderly population with perforated gastric ulcers. This may be explained by our study findings which indicate that elderly patients have a higher frailty, HLOS, and time to surgery than the adult population, with all three being a risk factor for mortality based on the present study. A study mentioned above, which looked at perforated gastric ulcers, found that the mean age of patients who died postoperatively was 13 years higher than those who survived [16]. Perforations aside, this information tells us that, postoperatively, younger patients tend to do better than older patients, reinforcing age as a critical risk factor for mortality in all gastric ulcer patients. This is seen in our data as the deceased population has an older age than the survived. Other authors looking at peptic ulcers (which factors in both gastric and duodenal ulcers as well as both perforated and non-perforated ulcers) also found age to be an important factor in overall survivability from peptic ulcers [2,17].

### 4.2. Ulcer Type and Mortality

Having considered age, ulcer type (hemorrhagic, perforated or both) becomes another important factor in mortality in these patients. In particular, gastric ulcer perforations come with a significantly higher morbidity and mortality than duodenal ulcer perforations, although much more infrequent [18,19]. Despite this difference, a recent study found that the presence of perforations was one of the most important risk factors for mortality in both adult and elderly populations, which is supported by the findings of this research [20]. Adults with a perforated ulcer or hemorrhagic perforated ulcer showed to be at higher risk of mortality, indicating that the presence of a perforations was indicative of mortality outcomes. This emphasizes the danger of gastric ulcer perforations. It has also been shown that when there is a hemorrhagic gastric ulcer, the first line treatment is commonly medical treatment; surgery is reserved for those who meet certain criteria [9]. Indeed, in the present study, of the patients who did *not* undergo surgery, over 95% of them were hemorrhagic ulcer patients; these hemorrhagic ulcer patients also had a mortality rate of 1.1% in the adult group and 2.2% in the elderly group, the lowest in the entire study. It becomes evident that hemorrhagic ulcers have a good prognosis compared to perforation, and surgery is included on a case-by-case basis. Furthermore, it has been shown that in elderly patients who have hemorrhagic gastric ulcers (in addition to severe ulcer disease and comorbidities) a nonsurgical approach, such as endoscopic hemostasis, was effective and safe [21]. On the other hand, the majority of perforated ulcer and hemorrhagic perforated ulcer patients fell into the category of patients requiring surgery, with very few opting for no surgery. This signified the necessity for surgery in patients with perforated gastric ulcers, regardless of age. Ultimately, physicians should remain cognizant in the management of gastric ulcer to maintain an index of suspicion for perforation to prevent poor outcomes. Evidently, some considerations must be made regarding elevated mortality risk in the elderly perforated ulcer patients, as they had the highest risk overall.

### 4.3. Surgery Status and Mortality

Mortality rates for surgical patients were statistically greater than their nonsurgical counterparts. This is a common theme among other surgical fields in the literature: the idea that surgery increases mortality rates compared to nonsurgical counterparts [22]. Our data show that the mortality rate was greater in all patients who underwent surgery compared to those who did not. Surgery also increases the mortality risk indirectly through increased HLOS [23]. According to the literature, the post-operative mortality of patients with perforated gastric ulcers [16,19,24,25,26] and perforated peptic ulcers [12,27] to be between ~10–20%, compared to <6% in the present study. A separate study found a 2.9% mortality in surgically treated patients with gastric ulcers, which was more in line with the findings of our study [28]. Interestingly, one randomized trial looking at perforated peptic ulcers found no differences in mortality between surgical and nonsurgical groups [7]. However, the same randomized trial also found that nonsurgical treatment failure was higher in patients over 70 years, meaning that surgery should be a serious consideration for elderly patients. Additionally, another study looking at perforated peptic ulcers and subsequent peritonitis over the course of nine years, found that their patients all underwent nonsurgical conservative medical treatments, and over 95% showed improvements and were discharged without operation [8]. Despite this variation in the literature, the present study suggests that surgery is a protective factor when considering the indices of severity in gastric ulcer perforations between surgical and nonsurgical groups, between both age groups. The results of our study clarified a few points: first, the presence of perforations in perforated ulcer and hemorrhagic perforated ulcer patients suggest these patients will undergo surgery, with overall lower mortality (<3% in adults; <6% in elderly) than what was represented in the literature [22]. Second, the absence of perforations in hemorrhagic ulcer patients suggest that these patients do not undergo surgery. Third, the presence of perforations was an important variable in the risk for mortality within gastric ulcer patients. Then, the more complicated patients had to be operated on, and the increased mortality in the surgical groups was a product of perforation complications and the surgical procedure itself. Meanwhile, nonsurgical and medical treatment methods were found to be superior for patients with hemorrhagic ulcers. These findings suggest that more work needs to be done to clarify specific surgical methods and whether their unique effects contribute to or have no effect on mortality in gastric ulcer patients.

### 4.4. Time to Surgery and Mortality

A significant contributing factor to mortality in patients with peptic ulcer perforations, either duodenal or gastric, is the time it takes to undergo surgery, or time to surgery. Time to surgery is a common risk factor in many surgical fields as well [13,15,20,22,23,29]. In one study looking at gastroduodenal perforations, patients that underwent surgery more than 12 h after initial perforation (pre-admission + time to surgery) had a much greater mortality than those who underwent surgery within the 12 h period. The authors concluded that the lower mortality was likely a result of less time for peritonitis to develop and more time for intestinal distention to manifest [10]. Interestingly, between the years 1935 and 1990, there was an increase in time to surgery for patients with perforated peptic ulcers, although the reasons are not clear as to why [4]. Other authors have corroborated the positive correlation between time to surgery and mortality for perforated peptic ulcers [17,30]. One study found that for every hour of delay in surgery for perforated peptic ulcers, there was a two to four percent decrease in probability of survival postoperatively [31]. This shows that any perforation in the upper gastrointestinal tract requires a short time to surgery, because of the high risk of mortality. However, it is important to consider that gastric ulcer perforations are two to three times more lethal than duodenal ulcer perforations [2]. Based on our data, perforated ulcer ulcers took the least amount of work-up and stabilization period before the operation as demonstrated by the shortest time to surgery. Additionally, hemorrhagic perforated ulcer patients took the longest time for the work-up and stabilization before the surgery, likely because of the added complications of hemorrhage. A marked delay in surgery for gastric perforated ulcers contributed to increased peritonitis spread and correlated with higher mortality than duodenal ulcers by a factor of four [32]. Therefore, a more focused look at gastric ulcers is required. Similar findings about surgical delay leading to increased mortality in gastric perforated ulcers were found in the literature [2,16,26,33]. This research points to the fact that a short time to surgery is critical for patient survivability, and the present study shows that indeed patients with gastric ulcer perforations all had the shortest time to surgery values for both age groups. A recent retrospective cohort study evaluated patients with complex abdominal wall reconstruction surgery and found that a shorter time to surgery led to a shorter HLOS, which can be predictive of decreased mortality as discussed below [34]. Additionally, our results showed that the surviving adult group had significantly shorter time to surgery values compared to the deceased group. The multivariable logistic regression model also found positive associations between mortality and increased time to surgery values in both adult and elderly groups. Early diagnosis and early treatment were found to be critical for a good prognosis in the elderly perforated gastric ulcer population [6]. All in all, the literature and the present study agree that patients with perforated peptic ulcers, especially gastric perforated ulcers, will derive more benefit and quality of life if the time it takes to get into surgery is minimized. Jordan et al. points out that, despite a shorter time to surgery correlating with lower mortality, it is important not to rush preoperative procedures in spite of getting the patient into surgery more quickly [10].

### 4.5. Hospital Length of Stay and Mortality

The mean HLOS of gastric perforated ulcers in both the adult and elderly groups in the present study loosely matched up with the literature: ~10 days [24]. However, in our study, perforated gastric ulcer and hemorrhagic perforated gastric ulcer patients both had significantly longer HLOS values than hemorrhagic gastric ulcer patients. Certainly, it is accepted that perforations are more deadly and have more complications than hemorrhages [6,9,33]. A separate study also found that elderly patients with perforated peptic ulcers stayed in the hospital longer than younger patients with the same condition, which is supported by our data [4]. These findings indicate that perforations are associated with more time spent in the hospital especially as patients’ age increases. Our study indicates that HLOS is higher in the surgical group for both age groups when compared to the non-surgical group, which can be explained by the combination of severity of gastric ulcers and complications and the surgical procedure that yield more time spent in the hospital. This longer HLOS by itself may also increase the odds of mortality [6,25]. In contradiction, one study found that, when comparing surgical and nonsurgical groups for patients with perforated peptic ulcers, HLOS was significantly increased in the *nonsurgical* group [7]. The authors concluded that this was likely due to nonsurgical treatment failure that led to complications and increased HLOS. This can help explain our findings as HLOS can be used as a predictor for mortality in adult and elderly patients who did not undergo surgery for gastric ulcer. In fact, in the general literature, HLOS is associated with mortality in surgical patients admitted emergently due to other medical conditions, such as bleeding gastritis, blunt force trauma, tracheostomy, etc., whereas one study found a reverse relationship between HLOS and mortality for emergently admitted patients with ruptured abdominal aortic aneurysms [35,36,37,38,39]. A few recent retrospective cohort studies analyzed patients with phlebitis/thrombophlebitis, acute pancreatitis, C. difficile colitis, and colon cancer, which found a significant increase in odds of mortality after day 4, 6, 6, and 15 of admission, respectively [40,41,42,43]. Our results also showed that the entire deceased population had a longer HLOS than that of the survived group. This is plausible because it is likely that patients with longer HLOS times were associated with increased complications that might have led to mortality. In a study looking at perforated gastric ulcers, it was found that the longer the elderly patients stayed in the hospital, the higher the rates of peritoneal infection spread and subsequent mortality and morbidity [16]. This may be why perforated gastric ulcer mortality in the elderly was so high in our study, likely because the elderly patients were more frail (higher modified frailty index) and spending more time in the hospital, which posed a greater risk for subsequent complications. This reinforces HLOS as one of the more important risk factors when considering mortality in gastric ulcer patients, regardless of ulcer type. In brief, as the literature for HLOS and mortality grows, actions to limit the number of days spent in the hospital are an important consideration in the treatment approach for patients with gastric ulcers. 

### 4.6. Comorbidities and Gastric Ulcers

Lastly, comorbidities have a very high degree of association with peptic ulcers. One study looking at perforated gastroduodenal ulcers found a significant association with end stage renal disease [36]. In the current study, a small percentage of patients in both age groups were associated with renal failure, although the elderly group had higher rates of comorbidity with this condition (>15%). Other forms of renal insufficiencies, liver failure and various forms of cancer were documented as significant comorbidities affecting mortality in patients with peptic ulcers [16,30,37]. A study on treatments for perforated gastric ulcers found that alcoholism played a significant role in the Boey risk score used in assessing mortality risk postoperatively [26]. The current study also found that many gastric ulcer patients were comorbid with alcoholism, with the adult group having higher associations than the elderly, as well as a significant male bias. Finally, gastric ulcers, unlike duodenal ulcers, are usually comorbid with cancer, with an incidence of about 6–14% [12]. Cancer is an important factor to consider because it plays a role in the type of surgery that is required [12,16]. Endoscopic procedures are very effective in correctly diagnosing peptic ulcers, and biopsies are important in ruling out malignancies, as these factor into the treatment plan [2]. In the current study, metastatic cancer, lymphomas, and solid tumors were comorbid with a small percentage of all patients, ranging from approximately 0–4%. In the clinical setting, sound diagnostic procedures must be implemented to rule out malignancies and allow for a successful treatment plan. Altogether, comorbidities account for more than half of the patients. These results align with values found in the literature, with ~50% of peptic ulcer patients having comorbidities [4,24]. Several authors also found significant positive correlations between comorbidity and mortality [16,17,26,27,30]. In particular, some authors have specifically found respiratory diseases, liver disease and cardiac disease as important comorbidities affecting their respective medical conditions [38,44]. Our study found these same comorbidities to be significant as well. It is also important to point out that fluid/electrolyte disorders and hypertension are the two most important comorbidities present in our study, which warrants further exploration of the associations between hypertension and fluid/electrolyte disorders with gastric ulcer patients.

### 4.7. Strengths and Weaknesses

There were a few limitations in the current study. First, the type of surgical procedure was not included. Several authors have pointed to the need for definitive surgery for the treatment of perforated gastric ulcers, due to the decreased mortality and fewer recurrences observed in some papers [18,45]. This makes the type of surgery an important consideration for mortality risk in these patients. Particularly, gastrectomy has been shown to be successful and protective of perforated gastric ulcers with a history of peptic ulcer disease, with low mortality rates and low recurrence of symptoms, especially when compared with simple closure [10,18]. Although peptic ulcer disease in the study indicates a chronic condition, there is little difference between acute and chronic, besides perseverance or relapse [46]. In fact, one study found that 79% of patients with perforated gastric ulcers had a prior history of peptic ulcer disease [26]. A similar study looking at perforated peptic ulcers found that a third of their patients had a history of peptic ulcer disease [8]. Although there are spontaneous gastric ulcers, differentiating acute from chronic is likely unnecessary. All in all, having surgery type in our study would have been very helpful in delineating specific effects of surgery on survivability in gastric ulcer patients. Second, since this is a retrospective study using a database, there may be confounding variables that were not accounted for in this research. Lastly, there is no data indicating postoperative complications, besides mortality. It would have been useful when discussing HLOS, since postoperative complications directly impact how long patients stay in the hospital after undergoing surgery for gastric ulcers. The main strength for this cohort study is the combination approach utilizing both the generalized additive model (GAM) and logistic regression model. Additionally, the sample size that was analyzed for this study was substantial. The nearly 7 million patient records yearly by the NIS database allows us to investigate across many domains while understanding patterns of care and disease outcome with a reliable sample size. Epidemiologic and demographic factors were considered in this study of acute gastric ulcer, which is scarce in the literature.

## 5. Conclusions

In conclusion, the results of this study suggest that surgery is a protective factor for patients who have a perforated ulcer, whether by itself or with a hemorrhage. Despite increases in mortality when compared with the nonsurgical group, the overall mortality of patients who underwent surgery was very good, compared to the literature. This is good news for patients with perforated gastric ulcers, as this is a very serious condition that often results in surgery. Those patients with only hemorrhagic ulcer are the more common type and most likely do not need surgery unless certain criteria are met. Lastly, a patient’s age can be a very important marker for the severity of a patient’s case.

## Figures and Tables

**Figure 1 ijerph-19-16263-f001:**
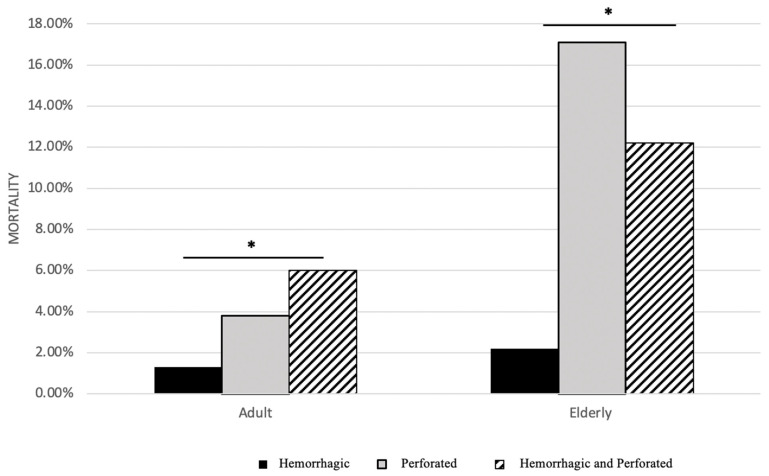
Overall Mortality. Mortality is plotted as a percentage, in relation to three subtypes of ulcers in both the adult group and the elderly group. The three subtypes of ulcers include: only hemorrhagic ulcer, perforated ulcer, and both. An asterisk (*) indicates significance (*p* < 0).

**Figure 2 ijerph-19-16263-f002:**
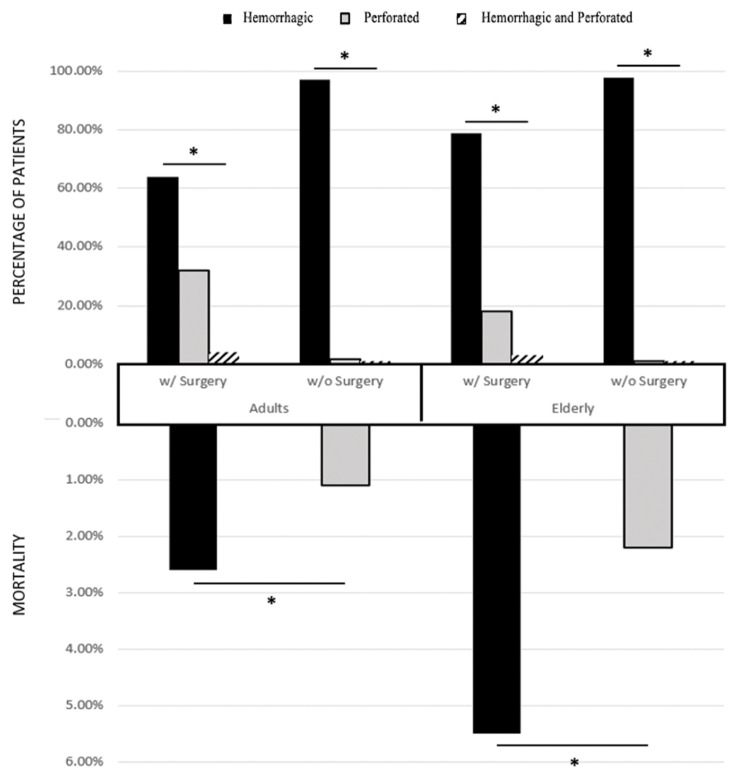
Comparing mortality with and without surgery. This is a comparison between the fraction of people electing or rejecting surgery with their subsequent mortality rates. The three subtypes of ulcers include: hemorrhagic ulcer, perforated ulcer, and both hemorrhagic perforated ulcers. An asterisk (*) indicates significance (*p* < 0).

**Figure 3 ijerph-19-16263-f003:**
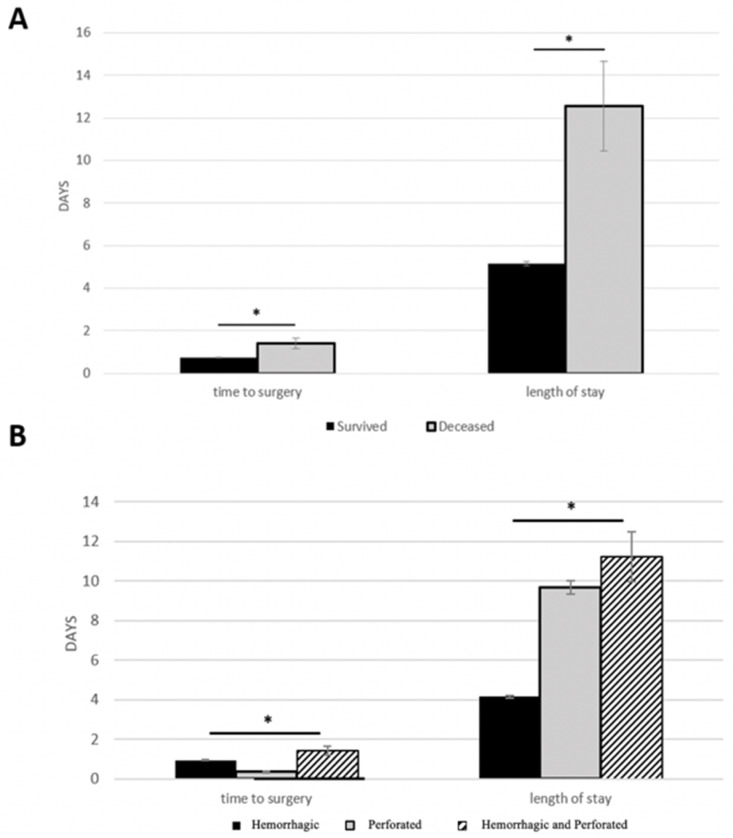
Time to surgery and Hospital Length of Stay (HLOS) in adults. Time to surgery and HLOS were plotted based on survival (**A**) and on ulcer type (**B**). An asterisk (*) indicates significance (*p* < 0.05). The three subtypes of ulcers include: hemorrhagic ulcer, perforated ulcer, and both hemorrhagic perforated ulcers. Standard error bars were calculated using standard deviation and *N* values from Table 2 and Table 4.

**Figure 4 ijerph-19-16263-f004:**
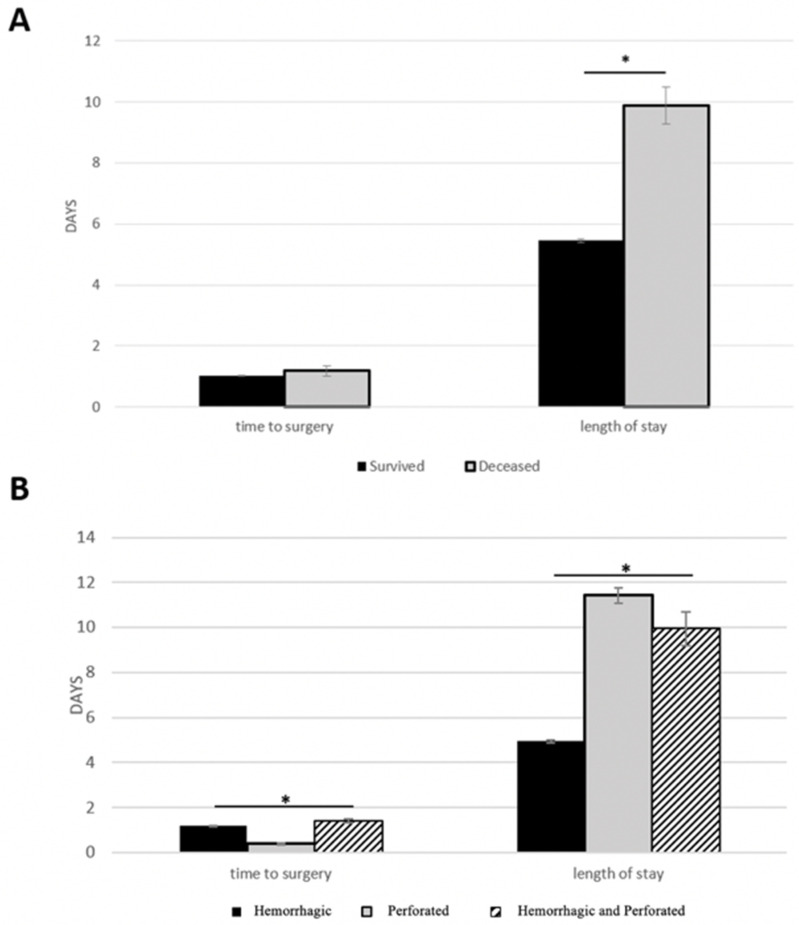
Time to Surgery and Hospital Length of Stay (HLOS) in elderly. Time to surgery and HLOS were plotted based on survival (**A**) and on ulcer type (**B**). An asterisk (*) indicates significance (*p* < 0.05). The three subtypes of ulcers include: hemorrhagic ulcer, perforated ulcer, and both hemorrhagic perforated ulcers.

**Table 1 ijerph-19-16263-t001:** The data presented in this table display the stratification of all emergently admitted gastric ulcer patients based on sex. Data were supplied by NIS 2005–2014.

		Adult, N (%)			Elderly, N (%)		
		Male	Female	*p*	Male	Female	*p*
All Cases		3658 (57.7%)	2679 (42.3%)		3863 (42.0%)	5334 (58.0%)	
Race	White	1748 (56.5%)	1604 (71.2%)	<0.001	2521 (75.5%)	3499 (78.9%)	0.004
	Black	607 (19.6%)	331 (14.7%)		297 (8.9%)	341 (7.7%)	
	Hispanic	388 (12.5%)	160 (7.1%)		228 (6.8%)	291 (6.6%)	
	Asian/Pacific Islander	198 (6.4%)	77 (3.4%)		202 (6.0%)	208 (4.7%)	
	Native American	33 (1.1%)	18 (0.8%)		7 (0.2%)	13 (0.3%)	
	Other	121 (3.9%)	63 (2.8%)		84 (2.5%)	84 (1.9%)	
Income Quartile	Quartile 1	1142 (32.3%)	738 (28.3%)	0.007	958 (25.6%)	1316 (25.1%)	0.460
	Quartile 2	923 (26.1%)	738 (28.3%)		967 (25.8%)	1430 (27.3%)	
	Quartile 3	785 (22.2%)	592 (22.7%)		908 (24.2%)	1257 (24.0%)	
	Quartile 4	682 (19.3%)	537 (20.6%)		915 (24.4%)	1239 (23.6%)	
Insurance	Private Insurance	1542 (42.4%)	1275 (47.7%)	<0.001	375 (9.7%)	354 (6.6%)	<0.001
	Medicare	575 (15.8%)	415 (15.5%)		3361 (87.1%)	4861 (91.2%)	
	Medicaid	600 (16.5%)	502 (18.8%)		59 (1.5%)	83 (1.6%)	
	Self-Pay	615 (16.9%)	335 (12.5%)		23 (0.6%)	15 (0.3%)	
	No Charge	54 (1.5%)	24 (0.9%)		2 (0.1%)	1 (0.0%)	
	Other	255 (7.0%)	121 (4.5%)		39 (1.0%)	14 (0.3%)	
Hospital Location	Rural	531 (14.5%)	448 (16.7%)	<0.001	635 (16.4%)	963 (18.1%)	0.009
	Urban: Non-Teaching	1560 (42.6%)	1202 (44.9%)		1750 (45.3%)	2485 (46.6%)	
	Urban: Teaching	1567 (42.8%)	1029 (38.4%)		1478 (38.3%)	1886 (35.4%)	
Comorbidities	AIDS	19 (0.5%)	5 (0.2%)	0.033	2 (0.1%)	0 (0%)	0.180
	Alcohol Abuse	815 (22.3%)	312 (11.6%)	<0.001	261 (6.8%)	130 (2.4%)	<0.001
	Deficiency Anemias	452 (12.4%)	372 (13.9%)	0.070	561 (14.5%)	808 (15.1%)	0.410
	Rheumatoid Arthritis	49 (1.3%)	122 (4.6%)	<0.001	78 (2.0%)	241 (4.5%)	<0.001
	Chronic Blood Loss	443 (12.1%)	356 (13.3%)	0.160	512 (13.3%)	801 (15.0%)	0.017
	Congestive Heart Failure	215 (5.9%)	139 (5.2%)	0.240	698 (18.1%)	969 (18.2%)	0.910
	Chronic Pulmonary Disease	410 (11.2%)	443 (16.5%)	<0.001	852 (22.1%)	1019 (19.1%)	<0.001
	Coagulopathy	308 (8.4%)	180 (6.7%)	0.012	318 (8.2%)	289 (5.4%)	<0.001
	Depression	212 (5.8%)	414 (15.5%)	<0.001	205 (5.3%)	507 (9.5%)	<0.001
	Diabetes, Uncomplicated	592 (16.2%)	335 (12.5%)	<0.001	947 (24.5%)	1026 (19.2%)	<0.001
	Diabetes, Chronic Complications	117 (3.2%)	92 (3.4%)	0.600	167 (4.3%)	201 (3.8%)	0.180
	Drug Abuse	277 (7.6%)	134 (5.0%)	<0.001	21 (0.5%)	22 (0.4%)	0.360
	Hypertension	1556 (42.5%)	1127 (42.1%)	0.710	2554 (66.1%)	3734 (70.0%)	<0.001
	Hypothyroidism	96 (2.6%)	258 (9.6%)	<0.001	263 (6.8%)	930 (17.4%)	<0.001
	Liver Disease	437 (11.9%)	237 (8.8%)	<0.001	172 (4.5%)	141 (2.6%)	<0.001
	Lymphoma	34 (0.9%)	16 (0.6%)	0.140	50 (1.3%)	39 (0.7%)	0.006
	Fluid/Electrolyte Disorders	991 (27.1%)	827 (30.9%)	0.001	1218 (31.5%)	1825 (34.2%)	0.007
	Metastatic Cancer	59 (1.6%)	44 (1.6%)	0.930	110 (2.8%)	81 (1.5%)	<0.001
	Other Neurological Disorders	167 (4.6%)	152 (5.7%)	0.046	262 (6.8%)	424 (7.9%)	0.036
	Obesity	294 (8.0%)	273 (10.2%)	0.003	196 (5.1%)	294 (5.5%)	0.360
	Paralysis	65 (1.8%)	39 (1.5%)	0.320	70 (1.8%)	106 (2.0%)	0.550
	Peripheral Vascular Disorders	121 (3.3%)	59 (2.2%)	0.009	385 (10.0%)	379 (7.1%)	<0.001
	Psychoses	137 (3.7%)	131 (4.9%)	0.025	65 (1.7%)	111 (2.1%)	0.170
	Pulmonary Circulation Disorders	30 (0.8%)	36 (1.3%)	0.043	115 (3.0%)	155 (2.9%)	0.840
	Renal Failure	299 (8.2%)	176 (6.6%)	0.017	674 (17.4%)	691 (13.0%)	<0.001
	Solid Tumor	63 (1.7%)	40 (1.5%)	0.480	129 (3.3%)	91 (1.7%)	<0.001
	Peptic Ulcer	8 (0.2%)	2 (0.1%)	0.210	0 (0%)	5 (0.1%)	0.060
	Valvular Disease	77 (2.1%)	91 (3.4%)	0.002	337 (8.7%)	492 (9.2%)	0.410
	Weight Loss	234 (6.4%)	165 (6.2%)	0.700	268 (6.9%)	369 (6.9%)	0.970
Gastric Ulcer	With Hemorrhage	2978 (81.4%)	2084 (77.8%)	0.002	3528 (91.3%)	4673 (87.6%)	<0.001
	With Perforation	588 (16.1%)	520 (19.4%)		268 (6.9%)	547 (10.3%)	
	With Hem and Perf	92 (2.5%)	75 (2.8%)		67 (1.7%)	114 (2.1%)	
Invasive Diagnostic Procedure		2478 (67.7%)	1797 (67.1%)	0.580	2855 (73.9%)	3928 (73.6%)	0.780
Surgical Procedure		1935 (52.9%)	1347 (50.3%)	0.039	1841 (47.7%)	2338 (43.8%)	<0.001
Invasive or Surgical Procedure		3550 (97.0%)	2591 (96.7%)	0.450	3764 (97.4%)	5174 (97.0%)	0.210
Deceased		74 (2.0%)	45 (1.7%)	0.320	151 (3.9%)	190 (3.6%)	0.390
		Mean (SD)	Mean (SD)	*p*	Mean (SD)	Mean (SD)	*p*
Age, Years		50.42 (10.65)	51.10 (10.35)	0.011	76.72 (7.50)	79.03 (7.80)	<0.001
Modified Frailty Index		1.04 (1.06)	1.01 (1.02)	0.240	1.69 (1.09)	1.57 (1.07)	<0.001
Time to Invasive Diagnostic Procedure, Days		1.09 (1.48)	1.17 (1.34)	0.090	1.34 (2.01)	1.39 (1.85)	0.270
Time to Surgery, Days		0.79 (1.56)	0.76 (1.66)	0.580	1.04 (1.82)	1.04 (1.97)	0.960
Hospital Length of Stay, Days		5.24 (7.60)	5.37 (6.80)	0.470	5.69 (6.05)	5.57 (5.90)	0.340
Total Charges, Dollars		40,573 (64,108)	42,512 (84,785)	0.310	41,125 (59,332)	37,823 (54,988)	0.007

**Table 2 ijerph-19-16263-t002:** The data presented in this table display the stratification of all emergently admitted gastric ulcer patients based on mortality. These data were supplied by NIS 2005–2014.

		Adult, N (%)			Elderly, N (%)		
		Survived	Deceased	*p*	Survived	Deceased	*p*
All Cases		6216 (98.1%)	119 (1.9%)		8857 (96.3%)	341 (3.7%)	
Sex, Female		2634 (42.4%)	45 (37.8%)	0.320	5143 (58.1%)	190 (55.7%)	0.390
Race	White	3286 (62.7%)	65 (62.5%)	0.980	5805 (77.5%)	215 (74.9%)	0.002
	Black	919 (17.5%)	19 (18.3%)		607 (8.1%)	32 (11.1%)	
	Hispanic	536 (10.2%)	11 (10.6%)		492 (6.6%)	26 (9.1%)	
	Asian/Pacific Islander	269 (5.1%)	6 (5.8%)		404 (5.4%)	6 (2.1%)	
	Native American	50 (1.0%)	1 (1.0%)		17 (0.2%)	3 (1.0%)	
	Other	182 (3.5%)	2 (1.9%)		163 (2.2%)	5 (1.7%)	
Income Quartile	Quartile 1	1843 (30.6%)	38 (33.3%)	0.650	2179 (25.2%)	94 (28.5%)	0.540
	Quartile 2	1633 (27.1%)	25 (21.9%)		2312 (26.7%)	85 (25.8%)	
	Quartile 3	1351 (22.4%)	26 (22.8%)		2087 (24.1%)	79 (23.9%)	
	Quartile 4	1194 (19.8%)	25 (21.9%)		2083 (24.1%)	72 (21.8%)	
Insurance	Private Insurance	2778 (44.9%)	37 (31.1%)	<0.001	708 (8.0%)	22 (6.5%)	0.870
	Medicare	970 (15.7%)	20 (16.8%)		7912 (89.4%)	309 (90.9%)	
	Medicaid	1061 (17.1%)	41 (34.5%)		138 (1.6%)	5 (1.5%)	
	Self-Pay	937 (15.1%)	13 (10.9%)		37 (0.4%)	1 (0.3%)	
	No Charge	78 (1.3%)	0 (0%)		3 (0.0%)	0 (0%)	
	Other	368 (5.9%)	8 (6.7%)		50 (0.6%)	3 (0.9%)	
Hospital Location	Rural	966 (15.5%)	12 (10.1%)	0.021	1544 (17.4%)	54 (15.8%)	0.410
	Urban: Non-Teaching	2718 (43.7%)	44 (37.0%)		4085 (46.1%)	151 (44.3%)	
	Urban: Teaching	2532 (40.7%)	63 (52.9%)		3228 (36.4%)	136 (39.9%)	
Comorbidities	AIDS	22 (0.4%)	2 (1.7%)	0.070	1 (0.0%)	1 (0.3%)	0.070
	Alcohol Abuse	1094 (17.6%)	33 (27.7%)	0.004	375 (4.2%)	16 (4.7%)	0.680
	Deficiency Anemias	812 (13.1%)	12 (10.1%)	0.340	1313 (14.8%)	55 (16.1%)	0.510
	Rheumatoid Arthritis	167 (2.7%)	4 (3.4%)	0.560	306 (3.5%)	13 (3.8%)	0.720
	Chronic Blood Loss	795 (12.8%)	4 (3.4%)	<0.001	1283 (14.5%)	30 (8.8%)	0.003
	Congestive Heart Failure	337 (5.4%)	17 (14.3%)	<0.001	1553 (17.5%)	114 (33.4%)	<0.001
	Chronic Pulmonary Disease	835 (13.4%)	18 (15.1%)	0.590	1777 (20.1%)	94 (27.6%)	<0.001
	Coagulopathy	447 (7.2%)	41 (34.5%)	<0.001	548 (6.2%)	59 (17.3%)	<0.001
	Depression	618 (9.9%)	7 (5.9%)	0.140	697 (7.9%)	14 (4.1%)	0.011
	Diabetes, Uncomplicated	910 (14.6%)	17 (14.3%)	0.910	1913 (21.6%)	60 (17.6%)	0.080
	Diabetes, Chronic Complications	207 (3.3%)	2 (1.7%)	0.320	358 (4.0%)	10 (2.9%)	0.310
	Drug Abuse	408 (6.6%)	3 (2.5%)	0.090	42 (0.5%)	1 (0.3%)	0.999
	Hypertension	2640 (42.5%)	43 (36.1%)	0.170	6109 (69.0%)	179 (52.5%)	<0.001
	Hypothyroidism	349 (5.6%)	5 (4.2%)	0.510	1168 (13.2%)	25 (7.3%)	0.002
	Liver Disease	636 (10.2%)	38 (31.9%)	<0.001	285 (3.2%)	28 (8.2%)	<0.001
	Lymphoma	48 (0.8%)	2 (1.7%)	0.240	85 (1.0%)	4 (1.2%)	0.570
	Fluid/Electrolyte Disorders	1749 (28.1%)	69 (58.0%)	<0.001	2838 (32.0%)	204 (59.8%)	<0.001
	Metastatic Cancer	94 (1.5%)	9 (7.6%)	<0.001	169 (1.9%)	22 (6.5%)	<0.001
	Other Neurological Disorders	309 (5.0%)	9 (7.6%)	0.200	650 (7.3%)	36 (10.6%)	0.026
	Obesity	560 (9.0%)	7 (5.9%)	0.240	481 (5.4%)	9 (2.6%)	0.024
	Paralysis	95 (1.5%)	9 (7.6%)	<0.001	165 (1.9%)	10 (2.9%)	0.160
	Peripheral Vascular Disorders	175 (2.8%)	5 (4.2%)	0.370	735 (8.3%)	29 (8.5%)	0.890
	Psychoses	261 (4.2%)	7 (4.9%)	0.370	170 (1.9%)	6 (1.8%)	0.830
	Pulmonary Circulation Disorders	63 (1.0%)	3 (2.5%)	0.130	257 (2.9%)	13 (3.8%)	0.330
	Renal Failure	456 (7.3%)	19 (16.0%)	<0.001	1293 (14.6%)	74 (21.7%)	<0.001
	Solid Tumor	99 (1.6%)	4 (3.4%)	0.130	204 (2.3%)	16 (4.7%)	0.005
	Peptic Ulcer	10 (0.2%)	0 (0%)	0.999	5 (0.1%)	0 (0%)	0.999
	Valvular Disease	165 (2.7%)	3 (2.5%)	0.999	795 (9.0%)	34 (10.0%)	0.530
	Weight Loss	378 (6.1%)	21 (17.6%)	<0.001	565 (6.4%)	72 (21.1%)	<0.001
Gastric Ulcer	With Hemorrhage	4992 (80.3%)	67 (56.3%)	<0.001	8022 (90.6%)	180 (52.8%)	<0.001
	With Perforation	1067 (17.2%)	42 (35.3%)		676 (7.6%)	139 (40.8%)	
	With Hem and Perf	157 (2.5%)	10 (8.4%)		159 (1.8%)	22 (6.5%)	
Invasive Diagnostic Procedure		4208 (67.7%)	64 (53.8%)	0.001	6621 (74.8%)	162 (47.5%)	<0.001
Surgical Procedure		3196 (51.4%)	86 (72.3%)	<0.001	3949 (44.6%)	232 (68.0%)	<0.001
Invasive or Surgical Procedure		6030 (97.0%)	109 (91.6%)	<0.001	8629 (97.4%)	309 (90.6%)	<0.001
		Mean (SD)	Mean (SD)	*p*	Mean (SD)	Mean (SD)	*p*
Age, Years		50.68 (10.55)	52.74 (8.97)	0.015	77.96 (7.72)	80.47 (8.34)	<0.001
Modified Frailty Index		1.02 (1.05)	1.45 (0.93)	<0.001	1.61 (1.08)	1.90 (1.16)	<0.001
Time to Invasive Diagnostic Procedure, Days		1.12 (1.42)	1.10 (1.39)	0.900	1.37 (1.91)	1.50 (2.50)	0.550
Time to Surgery, Days		0.76 (1.57)	1.41 (2.57)	0.036	1.03 (1.82)	1.18 (3.02)	0.470
Hospital Length of Stay, Days		5.16 (6.55)	12.56 (22.93)	<0.001	5.46 (5.60)	9.89 (11.18)	<0.001
Total Charges, Dollars		39,340 (68,828)	148,816 (171,455)	<0.001	36,886 (51,978)	98,745 (115,036)	<0.001

**Table 3 ijerph-19-16263-t003:** The data presented in this table display the stratification of all emergently admitted gastric ulcer patients based on operation/surgical status. These data were supplied by NIS 2005–2014.

		Adult, N (%)			Elderly, N (%)		
		No Operation	Operation	*p*	No Operation	Operation	*p*
All Cases		3055 (48.2%)	3283 (51.8%)		5019 (54.6%)	4181 (45.4%)	
Sex, Female		1332 (43.6%)	1347 (41.0%)	0.039	2996 (59.7%)	2338 (55.9%)	<0.001
Race	White	1564 (61.8%)	1788 (63.4%)	0.100	3323 (78.6%)	2698 (76.0%)	0.024
	Black	472 (18.7%)	466 (16.5%)		328 (7.8%)	311 (8.8%)	
	Hispanic	252 (10.0%)	296 (10.5%)		254 (6.0%)	265 (7.5%)	
	Asian/Pacific Islander	143 (5.7%)	132 (4.7%)		230 (5.4%)	180 (5.1%)	
	Native American	22 (0.9%)	29 (1.0%)		10 (0.2%)	10 (0.3%)	
	Other	77 (3.0%)	107 (3.8%)		82 (1.9%)	86 (2.4%)	
Income Quartile	Quartile 1	934 (31.6%)	947 (29.8%)	0.023	1265 (25.8%)	1009 (24.7%)	0.690
	Quartile 2	833 (28.2%)	828 (26.0%)		1308 (26.6%)	1090 (26.7%)	
	Quartile 3	632 (21.4%)	745 (23.4%)		1168 (23.8%)	998 (24.4%)	
	Quartile 4	560 (18.9%)	659 (20.7%)		1170 (23.8%)	985 (24.1%)	
Insurance	Private Insurance	1360 (44.7%)	1457 (44.6%)	0.110	370 (7.4%)	360 (8.6%)	0.100
	Medicare	516 (16.9%)	474 (14.5%)		4520 (90.2%)	3703 (88.7%)	
	Medicaid	514 (16.9%)	588 (18.0%)		80 (1.6%)	63 (1.5%)	
	Self-Pay	442 (14.5%)	509 (15.6%)		20 (0.4%)	18 (0.4%)	
	No Charge	34 (1.1%)	44 (1.3%)		1 (0.0%)	2 (0.0%)	
	Other	179 (5.9%)	197 (6.0%)		22 (0.4%)	31 (0.7%)	
Hospital Location	Rural	577 (18.9%)	402 (12.2%)	<0.001	1025 (20.4%)	573 (13.7%)	<0.001
	Urban: Non-Teaching	1271 (41.6%)	1491 (45.4%)		2222 (44.3%)	2015 (48.2%)	
	Urban: Teaching	1207 (39.5%)	1390 (42.3%)		1772 (35.3%)	1593 (38.1%)	
Comorbidities	AIDS	9 (0.3%)	15 (0.5%)	0.290	0 (0%)	2 (0.0%)	0.210
	Alcohol Abuse	515 (16.9%)	612 (18.6%)	0.060	183 (3.6%)	208 (5.0%)	0.002
	Deficiency Anemias	432 (14.1%)	392 (11.9%)	0.009	788 (15.7%)	581 (13.9%)	0.015
	Rheumatoid Arthritis	82 (2.7%)	89 (2.7%)	0.950	187 (3.7%)	132 (3.2%)	0.140
	Chronic Blood Loss	520 (17.0%)	279 (8.5%)	<0.001	823 (16.4%)	490 (11.7%)	<0.001
	Congestive Heart Failure	179 (5.9%)	175 (5.3%)	0.360	923 (18.4%)	745 (17.8%)	0.480
	Chronic Pulmonary Disease	374 (12.2%)	479 (14.6%)	0.006	950 (18.9%)	921 (22.0%)	<0.001
	Coagulopathy	202 (6.6%)	286 (8.7%)	0.002	256 (5.1%)	351 (8.4%)	<0.001
	Depression	315 (10.3%)	311 (9.5%)	0.260	414 (8.2%)	298 (7.1%)	0.045
	Diabetes, Uncomplicated	471 (15.4%)	456 (13.9%)	0.090	1080 (21.5%)	894 (21.4%)	0.870
	Diabetes, Chronic Complications	123 (4.0%)	86 (2.6%)	0.002	214 (4.3%)	154 (3.7%)	0.160
	Drug Abuse	170 (5.6%)	241 (7.3%)	0.004	21 (0.4%)	22 (0.5%)	0.450
	Hypertension	1334 (43.7%)	1349 (41.1%)	0.038	3481 (69.4%)	2808 (67.2%)	0.024
	Hypothyroidism	178 (5.8%)	176 (5.4%)	0.420	661 (13.2%)	532 (12.7%)	0.530
	Liver Disease	319 (10.4%)	355 (10.8%)	0.630	166 (3.3%)	147 (3.5%)	0.580
	Lymphoma	22 (0.7%)	28 (0.9%)	0.550	45 (0.9%)	44 (1.1%)	0.450
	Fluid/Electrolyte Disorders	797 (26.1%)	1021 (31.1%)	<0.001	1445 (28.8%)	1599 (38.2%)	<0.001
	Metastatic Cancer	33 (1.1%)	70 (2.1%)	<0.001	86 (1.7%)	105 (2.5%)	0.008
	Other Neurological Disorders	144 (4.7%)	175 (5.3%)	0.260	390 (7.8%)	296 (7.1%)	0.210
	Obesity	263 (8.6%)	304 (9.3%)	0.360	265 (5.3%)	226 (5.4%)	0.790
	Paralysis	49 (1.6%)	55 (1.7%)	0.820	100 (2.0%)	76 (1.8%)	0.540
	Peripheral Vascular Disorders	94 (3.1%)	86 (2.6%)	0.270	371 (7.4%)	393 (9.4%)	<0.001
	Psychoses	129 (4.2%)	139 (4.2%)	0.980	99 (2.0%)	77 (1.8%)	0.650
	Pulmonary Circulation Disorders	24 (0.8%)	42 (1.3%)	0.053	142 (2.8%)	128 (3.1%)	0.510
	Renal Failure	241 (7.9%)	234 (7.1%)	0.250	742 (14.8%)	625 (14.9%)	0.830
	Solid Tumor	47 (1.5%)	56 (1.7%)	0.600	106 (2.1%)	114 (2.7%)	0.060
	Peptic Ulcer	1 (0.0%)	9 (0.3%)	0.022	0 (0%)	5 (0.1%)	0.019
	Valvular Disease	86 (2.8%)	82 (2.5%)	0.430	473 (9.4%)	356 (8.5%)	0.130
	Weight Loss	137 (4.5%)	262 (8.0%)	<0.001	236 (4.7%)	401 (9.6%)	<0.001
Gastric Ulcer	With Hemorrhage	2965 (97.1%)	2097 (63.9%)	<0.001	4908 (97.8%)	3296 (78.8%)	<0.001
	With Perforation	55 (1.8%)	1054 (32.1%)		58 (1.2%)	757 (18.1%)	
	With Hem and Perf	35 (1.1%)	132 (4.0%)		53 (1.1%)	128 (3.1%)	
Invasive Diagnostic Procedure		2859 (93.6%)	1416 (43.1%)	<0.001	4759 (94.8%)	2026 (48.5%)	<0.001
Deceased		33 (1.1%)	86 (2.6%)	<0.001	109 (2.2%)	232 (5.5%)	<0.001
		Mean (SD)	Mean (SD)	*p*	Mean (SD)	Mean (SD)	*p*
Age, Years		51.20 (10.32)	50.26 (10.70)	<0.001	78.43 (7.80)	77.61 (7.68)	<0.001
Modified Frailty Index		1.02 (1.04)	1.02 (1.05)	0.900	1.59 (1.08)	1.66 (1.08)	0.004
Time to Invasive Diagnostic Procedure, Days		1.13 (1.14)	1.11 (1.89)	0.780	1.29 (1.31)	1.58 (2.93)	<0.001
Hospital Length of Stay, Days		3.55 (4.01)	6.93 (9.03)	<0.001	4.27 (3.38)	7.25 (7.73)	<0.001
Total Charges, Dollars		23,393 (25,979)	58,048 (96,070)	<0.001	25,671 (26,893)	55,477 (75,944)	<0.001

**Table 4 ijerph-19-16263-t004:** The data presented in this table display the stratification of all emergently admitted gastric ulcer patients based on ulcer type. These data were supplied by NIS 2005–2014.

		Adult				Elderly			
		Type of Ulcer Complication, N (%)			*p*	Type of Ulcer Complication, N (%)			*p*
		Hemorrhage	Perforation	Hem & Perf		Hemorrhage	Perforation	Hem & Perf	
All Cases		5062 (79.9%)	1109 (17.5%)	167 (2.6%)		8204 (89.2%)	815 (8.9%)	181 (2.0%)	
Sex, Female		2084 (41.2%)	520 (46.9%)	75 (44.9%)	0.002	4673 (57.0%)	547 (67.1%)	114 (63.0%)	<0.001
Race	White	2654 (61.9%)	610 (66.2%)	88 (62.9%)	0.060	5351 (77.1%)	550 (79.6%)	120 (81.6%)	0.190
	Black	744 (17.4%)	168 (18.2%)	26 (18.6%)		573 (8.3%)	55 (8.0%)	11 (7.5%)	
	Hispanic	468 (10.9%)	66 (7.2%)	14 (10.0%)		463 (6.7%)	51 (7.4%)	5 (3.4%)	
	Asian/Pacific Islander	234 (5.5%)	36 (3.9%)	5 (3.6%)		383 (5.5%)	21 (3.0%)	6 (4.1%)	
	Native American	41 (1.0%)	9 (1.0%)	1 (0.7%)		18 (0.3%)	2 (0.3%)	0 (0%)	
	Other	146 (3.4%)	23 (3.5%)	6 (4.3%)		151 (2.2%)	12 (1.7%)	5 (3.4%)	
Income Quartile	Quartile 1	1469 (30.0%)	356 (33.3%)	56 (34.1%)	0.120	2008 (25.0%)	219 (27.6%)	47 (26.9%)	0.370
	Quartile 2	1350 (27.5%)	264 (24.7%)	47 (28.7%)		2135 (26.6%)	218 (27.5%)	45 (25.7%)	
	Quartile 3	1090 (22.2%)	253 (23.6%)	34 (20.7%)		1957 (24.4%)	165 (20.8%)	44 (25.1%)	
	Quartile 4	995 (20.3%)	197 (18.4%)	27 (16.5%)		1925 (24.0%)	191 (24.1%)	39 (22.3%)	
Insurance	Private Insurance	2284 (45.3%)	468 (42.5%)	65 (39.2%)	<0.001	647 (7.9%)	72 (8.8%)	11 (6.1%)	0.950
	Medicare	833 (16.5%)	126 (11.4%)	31 (18.7%)		7337 (89.5%)	722 (88.7%)	164 (91.1%)	
	Medicaid	881 (17.5%)	188 (17.1%)	33 (19.9%)		127 (1.5%)	12 (1.5%)	4 (2.2%)	
	Self-Pay	703 (13.9%)	223 (20.3%)	25 (15.1%)		34 (0.4%)	3 (0.4%)	1 (0.6%)	
	No Charge	56 (1.1%)	20 (1.8%)	2 (1.2%)		3 (0.0%)	0 (0%)	0 (0%)	
	Other	290 (5.7%)	76 (6.9%)	10 (6.0%)		48 (0.6%)	5 (0.6%)	0 (0%)	
Hospital Location	Rural	789 (15.6%)	172 (15.5%)	18 (10.8%)	0.240	1421 (17.3%)	137 (16.8%)	40 (22.1%)	0.038
	Urban: Non-Teaching	2219 (43.8%)	462 (41.7%)	81 (48.5%)		3790 (46.2%)	356 (43.7%)	91 (50.3%)	
	Urban: Teaching	2054 (40.6%)	475 (42.8%)	68 (40.7%)		2993 (36.5%)	322 (39.5%)	50 (27.6%)	
Comorbidities	AIDS	20 (0.4%)	3 (0.3%)	1 (0.6%)	0.740	1 (0.0%)	1 (0.1%)	0 (0%)	0.120
	Alcohol Abuse	920 (18.2%)	166 (15.0%)	41 (24.6%)	0.003	339 (4.1%)	38 (4.7%)	14 (7.7%)	0.049
	Deficiency Anemias	686 (13.6%)	121 (10.9%)	17 (10.2%)	0.033	1186 (14.5%)	162 (19.9%)	21 (11.6%)	<0.001
	Rheumatoid Arthritis	141 (2.8%)	27 (2.4%)	3 (1.8%)	0.620	278 (3.4%)	35 (4.3%)	6 (3.3%)	0.400
	Chronic Blood Loss	765 (15.1%)	20 (1.8%)	14 (8.4%)	<0.001	1259 (15.3%)	26 (3.2%)	28 (15.5%)	<0.001
	Congestive Heart Failure	297 (5.9%)	47 (4.2%)	10 (6.0%)	0.100	1475 (18.0%)	157 (19.3%)	36 (19.9%)	0.550
	Chronic Pulmonary Disease	650 (12.8%)	184 (16.6%)	19 (11.4%)	0.003	1628 (19.8%)	210 (25.8%)	33 (18.2%)	<0.001
	Coagulopathy	423 (8.4%)	48 (4.3%)	17 (10.2%)	<0.001	532 (6.5%)	55 (6.7%)	20 (11.0%)	0.049
	Depression	520 (10.3%)	90 (8.1%)	16 (9.6%)	0.090	649 (7.9%)	45 (5.5%)	18 (9.9%)	0.028
	Diabetes, Uncomplicated	816 (16.1%)	97 (8.7%)	14 (8.4%)	<0.001	1819 (22.2%)	131 (16.1%)	24 (13.3%)	<0.001
	Diabetes, Chronic Complications	192 (3.8%)	14 (1.3%)	3 (1.8%)	<0.001	347 (4.2%)	19 (2.3%)	2 (1.1%)	0.004
	Drug Abuse	302 (6.0%)	90 (8.1%)	19 (11.4%)	0.001	33 (0.4%)	9 (1.1%)	1 (0.6%)	0.019
	Hypertension	2281 (45.1%)	343 (30.9%)	59 (35.3%)	<0.001	5672 (69.1%)	499 (61.2%)	118 (65.2%)	<0.001
	Hypothyroidism	298 (5.9%)	51 (4.6%)	5 (3.0%)	0.080	1071 (13.1%)	88 (10.8%)	34 (18.8%)	0.012
	Liver Disease	598 (11.8%)	56 (5.0%)	20 (12.0%)	<0.001	282 (3.4%)	22 (2.7%)	9 (5.0%)	0.270
	Lymphoma	36 (0.7%)	11 (1.0%)	3 (1.8%)	0.210	75 (0.9%)	11 (1.3%)	3 (1.7%)	0.300
	Fluid/Electrolyte Disorders	1410 (27.9%)	346 (31.2%)	62 (37.1%)	0.004	2565 (31.3%)	405 (49.7%)	74 (40.9%)	<0.001
	Metastatic Cancer	65 (1.3%)	36 (3.2%)	2 (1.2%)	<0.001	156 (1.9%)	30 (3.7%)	5 (2.8%)	0.003
	Other Neurological Disorders	256 (5.1%)	51 (4.6%)	12 (7.2%)	0.360	609 (7.4%)	64 (7.9%)	13 (7.2%)	0.900
	Obesity	481 (9.5%)	76 (6.9%)	10 (6.0%)	0.008	435 (5.3%)	45 (5.5%)	11 (6.1%)	0.870
	Paralysis	84 (1.7%)	18 (1.6%)	2 (1.2%)	0.900	164 (2.0%)	8 (1.0%)	4 (2.2%)	0.120
	Peripheral Vascular Disorders	158 (3.1%)	18 (1.6%)	4 (2.4%)	0.023	673 (8.2%)	75 (9.2%)	16 (8.8%)	0.590
	Psychoses	209 (4.1%)	49 (4.4%)	10 (6.0%)	0.470	151 (1.8%)	21 (2.6%)	4 (2.2%)	0.330
	Pulmonary Circulation Disorders	50 (1.0%)	14 (1.3%)	2 (1.2%)	0.700	246 (3.0%)	22 (2.7%)	2 (1.1%)	0.300
	Renal Failure	425 (8.4%)	35 (3.2%)	15 (9.0%)	<0.001	1257 (15.3%)	83 (10.2%)	27 (14.9%)	<0.001
	Solid Tumor	75 (1.5%)	21 (1.9%)	7 (4.2%)	0.018	186 (2.3%)	28 (3.4%)	6 (3.3%)	0.080
	Peptic Ulcer	8 (0.2%)	1 (0.1%)	1 (0.6%)	0.300	3 (0.0%)	1 (0.1%)	1 (0.6%)	0.009
	Valvular Disease	150 (3.0%)	14 (1.3%)	4 (2.4%)	0.006	766 (9.3%)	49 (6.0%)	14 (7.7%)	0.006
	Weight Loss	260 (5.1%)	114 (10.3%)	25 (15.0%)	<0.001	452 (5.5%)	152 (18.7%)	33 (18.2%)	<0.001
Invasive Diagnostic Procedure		3898 (77.0%)	286 (25.8%)	91 (54.5%)	<0.001	6466 (78.8%)	209 (25.6%)	110 (60.8%)	<0.001
Surgical Procedure		2097 (41.4%)	1054 (95.0%)	132 (79.0%)	<0.001	3296 (40.2%)	757 (92.9%)	128 (70.7%)	<0.001
Invasive or Surgical Procedure		4922 (97.2%)	1063 (95.9%)	157 (94.0%)	0.005	8002 (97.5%)	770 (94.5%)	168 (92.8%)	<0.001
Deceased		67 (1.3%)	42 (3.8%)	10 (6.0%)	<0.001	180 (2.2%)	139 (17.1%)	22 (12.2%)	<0.001
		Mean (SD)	Mean (SD)	Mean (SD)	*p*	Mean (SD)	Mean (SD)	Mean (SD)	*p*
Age, Years		51.37 (10.16)	47.90 (11.53)	49.44 (11.42)	<0.001	78.06 (7.72)	78.12 (8.22)	77.42 (7.42)	0.530
Modified Frailty Index		1.07 (1.05)	0.82 (0.98)	0.97 (1.02)	<0.001	1.62 (1.08)	1.61 (1.09)	1.59 (1.04)	0.870
Time to Invasive Diagnostic Procedure, Days		1.13 (1.21)	0.76 (2.44)	1.91 (3.75)	<0.001	1.35 (1.54)	2.01 (6.42)	1.66 (3.97)	<0.001
Time to Surgery, Days		0.95 (1.65)	0.36 (1.13)	1.42 (2.85)	<0.001	1.18 (1.96)	0.39 (1.12)	1.41 (3.12)	<0.001
Hospital Length of Stay, Days		4.15 (4.89)	9.66 (11.08)	11.23 (16.41)	<0.001	4.95 (4.92)	11.42 (9.81)	9.94 (9.45)	<0.001
Total Charges, Dollars		30,919 (43,185)	81,813 (133,980)	87,405 (112,660)	<0.001	32,867 (44,394)	93,098 (103,105)	85,172 (102,578)	<0.001

**Table 5 ijerph-19-16263-t005:** The outcome of the backward logistic regression analysis to evaluate associations between mortality, which was the dependent variable, and various factors is presented in this table. The groups analyzed here were only those that elected an operation/surgery. Data were supplied by NIS 2005–2014.

		Adult Operation		Elderly Operation	
		N = 2911	R^2^ = 0.061	N = 3696	R^2^ = 0.150
		OR (95% CI)	*p*	OR (95% CI)	*p*
Time to Surgery, Days		1.135 (1.046, 1.231)	0.002	1.100 (1.040, 1.163)	<0.001
Modified Frailty Index		1.520 (1.246, 1.854)	<0.001	1.203 (1.055, 1.372)	0.006
Gastric Ulcer	With Hemorrhage [Ref]		<0.001		<0.001
	With Perforation	2.802 (1.695, 4.630)	<0.001	7.535 (5.497, 10.329)	<0.001
	With Hem and Perf	2.931 (1.171, 7.342)	0.022	5.644 (3.051, 10.442)	<0.001
Age, Years		Removed Via Backward Elimination		1.052 (1.032, 1.072)	<0.001
Sex, Female				0.736 (0.545, 0.995)	0.047
Invasive Diagnostic Procedure				Removed Via Backward Elimination	
Race					
Income Quartile					
Insurance					
Hospital Location					

**Table 6 ijerph-19-16263-t006:** The outcome of the backward logistic regression analysis to evaluate associations between mortality, which was the dependent variable, and various factors is presented in this table. The groups analyzed here were only those that elected to not have an operation/surgery. Data were supplied by NIS 2005–2014.

		Adult Non-Operation		Elderly Non-Operation	
		N = 3053	R^2^ = 0.110	N = 5017	R^2^ = 0.159
		OR (95% CI)	*p*	OR (95% CI)	*p*
Hospital Length of Stay, Days		1.083 (1.039, 1.128)	<0.001	1.087 (1.052, 1.124)	<0.001
Invasive Diagnostic Procedure		0.153 (0.070, 0.334)	<0.001	0.240 (0.136, 0.424)	<0.001
Gastric Ulcer	With Hemorrhage [Ref]	Removed Via Backward Elimination			<0.001
	With Perforation			8.638 (4.126, 18.086)	<0.001
	With Hem and Perf			4.994 (1.954, 12.763)	<0.001
Age, Years				1.038 (1.012, 1.066)	0.004
Modified Frailty Index				1.369 (1.152, 1.628)	<0.001
Sex, Female				Removed Via Backward Elimination	
Race					
Income Quartile					
Insurance					
Hospital Location					

**Table 7 ijerph-19-16263-t007:** This table presents comorbidities of emergently admitted gastric ulcer patients, stratified according to survival status. The data were supplied by NIS 2005–2014.

	Adult, N (%)			Elderly, N (%)		
Comorbidities and Secondary Diagnoses (ICD-9 Codes)	Survived	Deceased	*p*-Value	Survived	Deceased	*p*-Value
Observations	6216 (98)	119 (2)		8857 (96)	341 (4)	
Tuberculosis (010.0–018.96)	1 (0.0)	0 (0)	0.999	0 (0)	0 (0)	
Bacterial Infections Other than Tuberculosis (020.0–041.9, 790.7)	980 (16)	48 (40)	<0.001	1247 (14)	143 (42)	<0.001
Nonbacterial Infections (042, 795.71, V08, 045.0–139.8, 790.8, and/or presence of Comorbidity of AIDS)	537 (9)	22 (19)	<0.001	325 (4)	19 (6)	0.070
Diabetes (250.0–250.93, V58.67, and/or presence of Comorbidity of Diabetes Uncomplicated or Diabetes Chronic Complications)	1118 (18)	19 (16)	0.570	2278 (26)	70 (21)	0.031
Hypertension (401.0–405.99, 796.2, and/or presence of Comorbidity of Hypertension)	2649 (43)	43 (36)	0.160	6115 (69)	179 (53)	<0.001
Anemia and/or Hemorrhage (280.0–285.9, 784.7, 784.8, and/or presence of Comorbidity of Anemia)	4324 (70)	71 (60)	0.020	7333 (83)	220 (65)	<0.001
Respiratory Diseases (415.0–417.9, 460–519.9, 784.91, 786, and/or presence of Comorbidity of COPD, ILD or Pulmonary Circulation Disease)	1557 (25)	92 (77)	<0.001	2919 (33)	255 (75)	<0.001
Coagulopathy (286.0–286.9, 790.92, V58.61, V58.63, and/or presence of Comorbidity of Coagulopathy)	684 (11)	45 (38)	<0.001	1458 (17)	70 (21)	0.048
Cardiac Diseases (391.X, 392.0, 393.398.99, 410.0–414.9, 420.0–429.9, 794.3X, 785.XX, and/or presence of Comorbidity of CHF or Valvular Diseases)	1884 (30)	83 (70)	<0.001	5243 (59)	293 (86)	<0.001
Cerebrovascular Diseases (325, 430–438)	128 (2)	11 (9)	<0.001	470 (5)	25 (7)	0.100
Peripheral Vascular Diseases (440–457.9, and/or presence of Comorbidity of Peripheral Vascular Disorders)	682 (11)	24 (20)	0.002	1473 (17)	55 (16)	0.810
Liver Diseases (570–573.9, 790.4, 794.8, and/or presence of Comorbidity of Liver Diseases)	748 (12)	46 (39)	<0.001	371 (4)	44 (13)	<0.001
Diseases of Digestive System other than Liver (530.00–569.9, 574.0–579.9, 787, 001.0–009.3, and/or presence of Comorbidity of Peptic Ulcer)	3928 (63)	62 (52)	0.013	5782 (65)	222 (65)	0.950
Diseases of Oral Cavity, Salivary Glands, and Jaws (520–529)	52 (1)	0 (0)	0.320	13 (0.1)	1 (0.3)	0.500
Nutritional/Weight Disorders (260–273.9, 275.XX,277.0–278.8, 783.XX, 799.3–799.4, and/or presence of Comorbidity of Weight Loss)	2106 (34)	44 (37)	0.480	4134 (47)	134 (39)	0.007
Endocrine Diseases (240.0–259.9, 991.0–992.9, and/or presence of Comorbidity of Endocrine Diseases)	1438 (23)	23 (19)	0.330	3242 (37)	90 (26)	<0.001
Genitourinary System Diseases (580.0–629.9, 403.XX, 791.XX, 788.XX, and/or presence of Comorbidity of Renal Diseases)	1390 (22)	75 (63)	<0.001	3543 (40)	220 (65)	<0.001
Neurological Diseases (317.0–326, 330.0–337.9, 340–359.9, 392, 780.0–780.09, 780.2–780.4, 317–319, 290.XX, 294.XX, 781.0–782.0, and/or presence of Comorbidity of Paralysis or Other Neurological Disorders or Paralysis)	1034 (17)	34 (29)	<0.001	2146 (24)	95 (28)	0.130
Diseases of the Musculoskeletal System and Connective Tissue (274.XX, 710.0–739, and/or presence of Comorbidity of Rheumatoid Arthritis or Lupus)	1393 (22)	18 (15)	0.060	3130 (35)	70 (21)	<0.001
Fluid and Electrolyte Disorders (275.0–276.9, 458.0–459.9, and/or presence of Comorbidity of Fluid and Electrolyte Disorders)	2258 (36)	78 (66)	<0.001	3523 (40)	224 (66)	<0.001
Neoplasms (140.0–239.9, V10.XX, and/or presence of Comorbidity of Lymphoma, Metastatic Diseases, or Tumor)	646 (10)	20 (17)	0.024	1793 (20)	70 (21)	0.900
Platelet and White Blood Cell Diseases (204.0–208.92, 287.0–288.9, 238.71)	584 (9)	21 (18)	0.002	729 (8)	56 (16)	<0.001
Psychiatric Diseases (293.XX, 295.0–302.9, 306.0–316, 780.1, V62.8, V15.4, and/or presence of Comorbidity of Psychoses)	1189 (19)	23 (19)	0.960	1189 (13)	35 (10)	0.090
Skin Diseases (680.0–709.9, 782.1–782.9)	312 (5)	11 (9)	0.038	583 (7)	41 (12)	<0.001
Trauma, Burns and Poisoning (800–999)	658 (11)	52 (44)	<0.001	786 (9)	156 (46)	<0.001
Drug Abuse/Withdrawal/Dependence (292.0–292.9, 304.0–304.93, 305.2–305.93, and/or presence of Comorbidity of Drug Abuse)	419 (7)	3 (3)	0.070	63 (0.7)	1 (0.3)	0.360
Alcohol Abuse/Withdrawal/Dependence (291.0–291.9, 303.0–303.93, 305.0–305.03, and/or presence of Comorbidity of Alcohol Abuse)	1094 (18)	33 (28)	0.004	375 (4)	16 (5)	0.680
Tobacco Use (305.1)	2065 (33)	28 (24)	0.026	1423 (16)	47 (14)	0.260
Long-Term Medications/Radiotherapy (V58.0–V58–2, V58.62, V58.64–V58.66, V58.68–V58.69)	569 (9)	2 (2)	0.005	1131 (13)	16 (5)	<0.001
Social Factors (V60.0–V62.6, V63.0–V64.3, V15.81)	211 (3)	6 (5)	0.330	143 (2)	2 (1)	0.140
Sleep Disorders (327, 780.5, V69.4, V69.5)	293 (5)	2 (2)	0.120	314 (4)	3 (1)	0.008
Lack of Physical Exercise (V69.0)	0 (0)	0 (0)		0 (0)	0 (0)	
Inappropriate Diet and Eating Habits (V69.1)	0 (0)	0 (0)		0 (0)	0 (0)	
High Risk Lifestyle Behaviors (V69.2, V69.3)	0 (0)	0 (0)		0 (0)	0 (0)	
Body Mass Index of Less than 18.9 (V85.0)	46 (15)	1 (50)	0.550	63 (21)	6 (50)	0.040
Body Mass Index of 19–24.9 (V85.1)	40 (13)	0 (0)		45 (15)	3 (25)	
Body Mass Index of 25.0–29.9 (V85.21–V85.25)	32 (11)	0 (0)		37 (12)	0 (0)	
Body Mass Index of 30.0 and over (V85.30–V85.45)	187 (61)	1 (50)		156 (52)	3 (25)	

## Data Availability

Data will be available upon request.

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
