# Peer review of "Risk Factors for Mortality in Emergently Admitted Patients with Acute Gastric Ulcer: An Analysis of 15,538 Patients in National Inpatient Sample, 2005–2014"

_ijerph, 2022, doi:10.3390/ijerph192316263_

Round 1
Reviewer 1 Report
Dear authors:
Please find my comments below:
1. The title is biased and not reflecting the content. Those geriatric patients with peptic ulcers definitely have higher risk compared with adult patient. This is true even not looking at the statistical analyses. This has to be improved.
2. Abstract - it is written as p-value and not P-value. All patients included in the analyses or selected patients only? Total 15, 538 is the total or selective? Please make it clear what is the boundary for age groups. Adult is 18 to 65 years? Geriatric is more than 65?
3. References missing in the few sentences in the introduction.
4. Materials and Methods - What does it mean by standard t-test? Please be clear.
5. Results - some values/data were repetitive in the abstract section. p-value was significant for race and how to interpret this? Also income quartile and insurance. The same for hospital location and co-morbidities etc. Please add caption details for the tables. Figure not so good. Please improve the clarity.
6. Discussion - Please add the 'strengths and weaknesses' section in the discussion section.
Thank you
Author Response
- The title is biased and not reflecting the content. Those geriatric patients with peptic ulcers definitely have higher risk compared with adult patient. This is true even not looking at the statistical analyses. This has to be improved.
- Thank you for this feedback, we believe you bring up a good point. We have changed our title to:
- “Risk factors of mortality in patients with acute gastric ulcer admitted emergently: an analysis of 15,538 patients in NIS”
- Abstract - it is written as p-value and not P-value. All patients included in the analyses or selected patients only? Total 15, 538 is the total or selective? Please make it clear what is the boundary for age groups. Adult is 18 to 65 years? Geriatric is more than 65?
- Thank you for your points. We have adjusted our P-values as p-values and have clarified that we used a total of 15,538 patients in the study.
- Yes, you are right. Adults are 18-64 years old and elderly patients are 65 and above in our study, this is mentioned in line 18 for the abstract and again on line 89 of the methods.
- References missing in the few sentences in the introduction.
- We appreciate this suggestion. We have adjusted the references in the first few sentences of the introduction.
- Materials and Methods - What does it mean by standard t-test? Please be clear.
- Thank you for this referencing this and helping us improve our paper. We have clarified the standard t-test with greater detail in the methods section.
- Results - some values/data were repetitive in the abstract section. p-value was significant for race and how to interpret this? Also income quartile and insurance. The same for hospital location and co-morbidities etc. Please add caption details for the tables. Figure not so good. Please improve the clarity.
- Thank you for these suggestions. We have realized that the wording in the sentences regarding race, income quartile, insurance, and hospital location that you’re referring to was misleading, and have edited them. We have also adjusted the captions for the tables and enlarged the figures for clarity.
- Discussion - Please add the 'strengths and weaknesses' section in the discussion section.
- We are grateful for this suggestion. We have improved upon our Strengths and Weaknesses section in the Discussion section.
Reviewer 2 Report
The current study examines risk factors for gastric ulcer mortality in adult and geriatric patients.
The Paper is well written with a good flow but some of the queries need to be addressed by the authors.
1. National database is not transparent, more details are to be included on the area covered and total population. In the methodology section, appropriate sample size calculations are required.
2. It's no wonder that age is the most important risk factor for mortality in your study. The authors have widely classified the age categories such as (18-64-year-old) adults and (65-year-old) elderly; however, it would have been more fascinating if the authors had precisely classified the elder age, and then this research might have revealed the exact risk factors along with age. It is evident that a 100-year-old patient is more likely to die from a gastric ulcer than a 65-year-old patient, regardless of other comorbid conditions.
Author Response
- National database is not transparent, more details are to be included on the area covered and total population. In the methodology section, appropriate sample size calculations are required.
- Thank you for your suggestions. We agree that we weren’t completely transparent regarding the database used for our study. Thus, we have improved upon this information in the methods section.
- Regarding the sample size, we included all the population available from the database that met the inclusion criteria.
- It's no wonder that age is the most important risk factor for mortality in your study. The authors have widely classified the age categories such as (18-64-year-old) adults and (65-year-old) elderly; however, it would have been more fascinating if the authors had precisely classified the elder age, and then this research might have revealed the exact risk factors along with age. It is evident that a 100-year-old patient is more likely to die from a gastric ulcer than a 65-year-old patient, regardless of other comorbid conditions.
- Thank you for this comment. You bring up a very valuable point that we can further clarify in the paper. Specifically, we lose information if we were to categorize continuous variables. Therefore, age was used as a continuous variable that was not categorized in this regression model, which provides deeper information about the relationship of age and mortality for patients admitted emergent with gastric ulcers.
- For example, in table 5, there is about a 5% increased odds of mortality in operative patients with each additional year of age among elderly patients. Similarly, there is a 4% increased odds of mortality in non-operative patients with each additional year of age among elderly patients. Interestingly, age did not emerge as a risk factor of mortality for operative and non-operative groups among adult patients. I think you agree with us that age as a continuous variable reflects better its relationship with mortality. For example, an 85 year old has about a 100% higher odds of mortality than a 65 year old (about 2 times higher), based on the fact that elderly patients experience a 5% higher risk of mortality for each additional year of age.
- This concept was clarified to the results section of the paper.